# No Apparent Immediate Reproductive Costs of Overlapping Breeding and Moult in a Mediterranean Great Tit Population

**DOI:** 10.3390/ani13030409

**Published:** 2023-01-26

**Authors:** Iris Solís, Elena Álvarez, Emilio Barba

**Affiliations:** Cavanilles Institute of Biodiversity and Evolutionary Biology, University of Valencia. c/ Catedrático José Beltrán 2, 46980 Paterna, Spain

**Keywords:** breeding–moult overlap, clutch size, fledgling production, *Parus major*, resource allocation, Spain

## Abstract

**Simple Summary:**

The annual breeding and moulting calendar in birds is affected by global warming and is leading to an increased overlap between these two activities in some populations. Allocation of resources for two energy-demanding activities could negatively affect performance in one or both of them. In this paper, we examined whether the overlapping of breeding and moulting has negative effects on the reproductive performance of a population of great tit (*Parus major*) in eastern Spain. We found that, in pairs where both parents overlapped breeding and moulting, clutch size was smaller, fewer eggs hatched, and fewer fledglings in poorer body condition left the nest. However, these differences disappeared when the seasonal trend in breeding performance was taken into account, i.e., the poorer reproductive performance of pairs that overlap moulting and breeding was mainly due to the fact that they breed later, and reproductive performance declines as the season progresses. Thus, we conclude that the overlap of breeding and moulting does not impose additional immediate reproductive costs in this population.

**Abstract:**

Some phenological events in birds, such as breeding and moulting, are being affected by rising temperatures due to global warming, and many species have undergone temporary changes in these energetically demanding phases that are often separated in time. This has led to an increased overlap between breeding and moulting in some populations. This overlap causes conflicts in resource allocation and may impose fitness costs that could affect immediate reproductive performance. We tested whether this occurs in a great tit (*Parus major*) population in eastern Spain. In 71% of 390 pairs, in which both parents were captured during the period of overlap between moulting and breeding, at least one parent was moulting when feeding the chicks of its second brood. Later breeders were more likely to overlap breeding and moulting, and when both parents overlapped, clutch size was smaller, fewer eggs hatched and fewer fledglings in poorer body condition were produced. Some results were intermediate when only one parent moulted. However, all these differences between moulting and non-moulting pairs disappeared when the seasonal trend in reproductive parameters was taken into account, as moulting birds bred later and reproductive performance decreased seasonally. Therefore, the overlap of breeding and moulting does not impose additional reproductive costs in this population.

## 1. Introduction

The increase in temperatures during the last decades due to global warming is affecting phenological events in many species of plants and animals [1,2,3,4,5,6,7]. Reproduction, moult and migration are some examples of phenological events that have been affected by global warming, and birds undergo through these important life stages in a specific sequence. Each stage is energetically costly, and they tend to avoid overlapping them [8,9,10,11]. In many bird species, the timing of reproduction, moult and/or migration is altered [12,13,14,15,16,17,18]. Moreover, the relative temporal changes in the duration of these stages differ in different rates, so that the periods elapsed between them are either lengthened or shortened [17]. In some extreme cases, intervals between activities can even disappear, thus two of them (as reproduction and moult) could overlap [17,18].

Overlapping breeding and moulting leads to a conflict in resource allocation between these two energy-demanding processes. This can lead to physiological trade-offs that can include a reduction in the breeding success, survival rate and/or the quality of the feathers [10,17]. However, probably because of its rare occurrence in normal conditions, few studies have explored the reproductive consequences of moult–breeding overlap. Moreover, these few studies have been mostly performed on migratory species, where the pressure to finish moulting before migration is higher. Most of these studies adopt different experimental approaches [10,19,20,21,22,23,24], while only few of them were based on observation of non-manipulated individuals [17,25,26,27]. Experimental studies are good for showing whether forcing the birds to overlap when they would not have done so caused overlapping birds to experience negative fitness effects. However, it is important to know whether this also happens in real (i.e., non-manipulated) conditions. It might well be that overlapping birds are in better body condition, had better territories with more available food, and/or are supporting lower reproductive burdens (i.e., lower brood size); and so they might efficiently perform both activities without apparent costs. Therefore, observational studies are important to shed light on the consequences of overlapping for non-manipulated individuals.

All studies including observations in unmanipulated conditions have been performed with the migratory pied flycatcher (*Fidecula hypoleuca*), each one with a different approach and measuring different variables in different ways. In an initial study, Hemborg (1999) [25] included five years of observations and 96 complete pairs (i.e., both male and female captured). He therefore compared some reproductive parameters (laying date, clutch size, number of fledglings, and fledging success) of pairs in which none; only the male; only the female; or both were moulting. Then, Hemborg et al., (2001) [26] built over the previous study, by adding data of three more European flycatcher populations. In this case, they only included in the analyses laying date, clutch size and breeding success, and males and females were treated independently (i.e., not considering pairs as a whole). Morales et al. (2007) [27] reported on the effects of overlap on hatching success and on hatchlings and fledglings produced by females. Finally, Tomotani et al., (2017) [17] performed a study with a big dataset of 35 years to study temporal changes in the timing of breeding and moulting, and the impact of their overlap on clutch size, proportion of chicks fledged and adult survival. Overall, these studies detected negative consequences of overlapping on breeding performance, but there were large differences between them in which traits were affected.

The aim of this study was to investigate the reproductive consequences of moult–breeding overlap in a Mediterranean great tit (*Parus major*) population, where the proportion of adults overlapping both activities have increased throughout the years [18]. This has provided the opportunity to reach a reasonable sample size of naturally overlapping individuals. Thus, this is the first observational study dealing with the potential cost of reproduction of moult-breeding overlap in a non-migrant bird. We also worked with a notably high sample size in a single population (390 complete pairs), so the consequences of different combinations (only the female, only the male, both, or neither overlapping) could be compared. Based on previous studies, and on theoretical background [28,29], we would expect that pairs in which at least one member of the pair is overlapping would show smaller clutches, lower number of hatchlings and fledglings, and produce fledglings with poorer body condition. 

## 2. Materials and Methods

This study was performed between 1995 and 2019 on a resident wild population of great tits breeding in nest boxes placed in an extensive orange (*Citrus aurantium*) monoculture near Sagunto (Valencia, eastern Spain; 39°42′N, 0°15′W, 30 m a.s.l.). The size of the study area has increased over the years, from approximately 150 ha to about 450 ha in recent years. The density of nest boxes has remained constant at 1 nest box per ha. The type and structure of the habitat have not changed along these years. However, the increasing length of spring means ambient temperatures throughout the study period caused and advancement of the breeding season and changes in the seasonal distribution of clutches [18], which is probably the cause of the increased degree of moult–breeding overlap [30].

Each nest box was inspected weekly throughout the breeding season (March to June), and daily by the time of expected clutch completion, hatching and fledgling dates, to record basic breeding parameters [18,31], such as laying date, clutch size and number of hatchlings and fledglings. We considered the laying date as the day the first egg was laid, assuming that one egg was laid per day. Dates are always presented as “April dates” (i.e., 1 April = Day 1). Laying was assumed to have finished when no more eggs were laid during two consecutive days and the female started full incubation. As parents remove dead small nestlings, those disappearing shortly after hatching were considered dead. Nestlings were individually ringed, their tarsus length measured (digital caliper to the nearest 0.1 mm), and their body mass recorded (digital balance to the nearest 0.1 g) when they were 15 days old. Then, the body condition of each nestling was computed as the body mass to tarsus length ratio [31,32]. Mean tarsus length, body mass and body condition of the nestlings of each nest were computed and used in the analyses. Nests were visited after the expected fledging date to look for dead nestlings and determine the number of fledglings. Parents are not able to remove large dead nestlings [33], so those dead by starvation remained in the nest. On the other hand, predators active in the study area kill and usually eat all the nestlings present. Mammals, such as weasels *(Mustela nivalis*) [34], garden dormouse (*Eliomys quercinus*) [35], or black rats (*Rattus rattus*) [36] leave different remains, usually including legs with the metal rings. The Montpellier snake (*Malpolon monspessulanus*) sometimes visits nests, swallowing nestlings whole and leaving no remains in the nest [37]. However, it usually acts on smaller (i.e., before ringing) nestlings and, in any case, the nest is left very flat; a somewhat different state as when nestlings fledge. In almost 30 years ringing nestlings in the study area, we have never recovered a bird alive after it was classified as “dead in the nest” due to its disappearance after an act of predation.

Parents were captured with door traps at the nestbox when nestlings were 10–12 days old. They were ringed with individually numbered metal rings if not already ringed. Sex and age (yearling or older) were recorded [38] and it was noted whether the bird was moulting its primary feathers or not [27]. Adult great tits have a complete postnuptial moult and they rarely moult feathers other than their primaries while breeding [39]. In our case, the most advanced birds were moulting their fourth primary while feeding 10–12-day-old nestlings. All the birds included in this study were raising their second (after a successful clutch) or replacement (after a failure of the first one) clutch; none of the individuals raising their first brood were found moulting [18].

Our aim in this study was to determine the reproductive consequences of overlapping breeding and moult, so we have only used data from years when there was at least one adult moulting while breeding. For each year, we considered that moult started the first day we caught the first adult moulting at least one of its primary wing feathers. No bird captured while raising its first brood was found moulting in any of the study years, so all the birds included here were actually attempting a second clutch, whether the first one had been successful or not. Birds attempting only one clutch could have started moult before our estimated “date of moult start” [39,40], but this does not affect our analyses since none of these birds overlapped breeding and moult. Thus, the date of laying of the first egg of the second clutch of the pair in which one of the components was first found moulting each year was taken as a benchmark, and all the pairs that started laying at this date or later were considered potential overlapping pairs for that year. These will be named “late-breeding pairs” hereafter.

Considering the above restriction, our dataset was split into four groups: nests where none of the parents were overlapping breeding and moult, nests where only the male was overlapping, nests where only the female was overlapping, and nests where both parents were overlapping. Only pairs where both parents were captured were included. To analyze the effect of overlapping on each variable, we used all the pairs for which the value of this particular variable was known; therefore, sample size might differ between analyses. As we will detail in the Results section, we used 390 complete pairs, in which male and female were individually identified, for our analyses. Of these pairs, only 33 pairs were considered in two (31 pairs) or three years (2 pairs), while 322 (83%) were considered only once. Additionally, 4 males were considered twice, and 8 were considered three times paired with different females; while 5 females were considered twice, and 7 were considered three times paired with different males. As these are relatively small numbers, the potential bias due to pseudo-replication would be very low, so we avoided performing more complex analyses (as mixed models). 

In order to remove the effect of the year in our analyses, actual values for all the variables were transformed into z-scores by subtracting the mean and dividing by the standard deviation of that variable for that year. To have robust estimates of these statistics, we eliminated years in which fewer than 16 pairs could have overlapped breeding and moulting. Differences in reproductive parameters between the four types of pairs were tested using ANOVAs, followed by a posteriori Tukey’s HSD when significant. In addition, as reproductive performance used to decrease through the season, all the analyses were repeated as ANCOVAs, using the z-values of laying dates as a covariate to distinguish the potential effects of overlapping moult and breeding from that of seasonal variation. 

All analyses were conducted with the IBM SPSS Statistics 25 software

## 3. Results

### 3.1. Incidence of Moult-Breeding Overlap

Given their breeding and moult start dates for each specific year, from 1995 to 2019, individuals from a total of 564 pairs could have overlapped breeding and moulting. From these, 390 complete pairs were captured while feeding nestlings. In 29.0 % of these pairs, none of the birds were moulting; both parents were moulting in 20.8 % of the cases; only the male in 45.6 %; and only the female in 4.6 % of the pairs. In other words, at least one member of the pair overlapped moult and reproduction in 71% of late-breeding pairs.

Considering only years in which at least 16 pairs were captured during the period of potential moult–breeding overlap, those pairs in which both members overlapped moult and breeding started laying later than the rest of the groups (ANOVA: F_1, 333_ = 28.378; *p* < 0.001, and Tukey post-hoc pairwise comparisons; Figure 1).

### 3.2. Reproductive Consequences of Moult–Breeding Overlap

Clutch size differed between groups, being smaller when both parents overlapped than when only the male or neither parent did, while those cases in which only the female overlapped were intermediate, not differing from any or the other groups (ANOVA: F_1,331_ = 9.042; *p* < 0.001, and Tukey tests). However, these differences disappeared when laying date was added as a covariate into the analysis (ANCOVA, moult group: F_3,331_ = 2.537, *p* = 0.057; laying date: F_1,331_ = 41.130, *p* < 0.001; Figure 2).

Results for clutch size were exactly the same for the number of hatchlings both when comparing directly the four types of pairs (ANOVA: F_1,332_ = 5.208; *p* = 0.002, and Tukey tests) and when laying date was added as a covariate (ANCOVA, moult group: F_3,331_ = 1.339, *p* = 0.262; laying date: F_1,332_ = 25.631, *p* < 0.001).

The number of fledglings produced was clearly lower for pairs where both parents overlapped than for those where none of them did it, being intermediate for those in which only one of the parents (either male or female) overlapped (ANOVA: F_1,333_ = 3.217; *p* = 0.023, and Tukey tests) but, again, the differences disappeared when laying date was included as a covariate (ANCOVA, moult group: F_3,331_ = 0.505, *p* = 0.679; laying date: F_1,332_ = 35.570, *p* < 0.001; Figure 3).

Finally, the nestlings showed a poorer body condition if both parents overlapped than if none, or only the male did it; the body condition was intermediate if only the female was moulting while attending them (ANOVA: F_1,304_ = 4.532; *p* = 0.004; Tukey test), but this effect among groups disappeared when we included laying date as a covariate (ANCOVA, moult group F_3,302_ = 0.792; *p* = 0.499; laying date: F_1,302_ = 21.799, *p* < 0.001).

## 4. Discussion

It is well-known that males tend to start moulting earlier than females, and thus they have been found to overlap these two activities more frequently [18,25,39,41,42]. The usual explanation is that females are more energy-limited during the first stages of breeding, as they spend much more time incubating or brooding the hatchlings [11]. In our sample of 390 pairs, 259 (66%) males were moulting, while only 99 (25%) females overlapped breeding and moult. This should be considered as a conservative figure, since parents were trapped when nestlings were 10–12 days old, so individuals starting moulting during the second half of the nestling period remained undetected.

An important difference of our study with previous observational ones is that great tits are resident, i.e., do not have the pressure to moult early or quickly in order to migrate. We have previously shown that, in the studied population, the increase of pairs laying two clutches, probably due to recent ambient temperature increase, has also increased the number of pairs overlapping breeding and moult [18]. As presented above, results from both observational and experimental studies led us to expect a reproductive cost of such overlap of two energy-demanding activities. Our main question in this study was to test this expectation. We actually found that overlapping pairs showed a poorer breeding performance than non-overlapping ones, with generally intermediate scores for pairs in which only one of the members overlapped. However, this reduced performance is mostly explained by the seasonal decreasing trend. As overlapping breeding and moult was more frequent as the season progressed, pairs breeding late overlapped more frequently and showed reduced breeding performance, but there was not an observable additional cost of the moult–breeding overlap. We can only conclude from the available data that individuals overlap breeding and moulting if they can afford to do so, without imposing additional costs on their current breeding. We will discuss in turn the studied breeding parameters.

The first one that could be negatively affected by allocating resources to moult is clutch size. Although we declared that, based on previous studies, our expectation was that clutch size would be smaller in pairs where overlap occurred, the details of these studies merit a closer look. There are, to the best of our knowledge, two studies dealing with the effect of overlapping on clutch size, both in pied flycatchers. The first study [25] gave a clear result for a Swedish population: pairs where both parents were moulting showed smaller clutches than those in which only the male, or none of the parents, were moulting; those in which only the female was moulting showed and intermediate value, suggesting that clutch size was more affected if females were overlapping. These differences were maintained when the effects of the seasonal decrease in clutch size were removed. The second study [26] is more difficult to interpret, since it did not consider pairs but individuals, separately analyzing males and females. In none of the four European populations considered (including the Swedish one mentioned above) were differences in clutch size found between overlapping and non-overlapping females. The results with males were more striking, since in two populations clutch size was smaller in nests attended by moulting males, while in the other two the opposite was true. However, these differences disappeared when laying date was considered. We should observe that in both studies [25,26], the population of Abisko (Sweden) was included and the same data used, with totally opposite results (negative effects of overlapping in the first, no effect in the second). This difference could be only attributed to the way the data were analysed. Our results with great tits generally agree with those of Hemborg et al. (2001) [26], in that clutch size was not directly affected by moult status. In our case, if both male and female were overlapping, clutch size was lower, but this reduction of clutch size could be attributed to the later breeding dates of these pairs.

We found that the number of hatchlings produced by the different groups of pairs mirrored those of clutch size, indicating that there was no effect on incubation efficiency. The only study allowing a sensible comparison is that of Hemborg (1998) [20], an experimental study where breeding time was either advanced or delayed by exchanging clutches during incubation. In this study, clutch size could obviously not be affected, and hatching success was similar between overlapping and non-overlapping pairs. Results of this study were, therefore, in agreement with those presented here for Spanish great tits, suggesting that overlapping incubation and moult does not affect female incubation efficiency. This reinforces the idea that the effect of overlap on clutch size would be low, as the incubation period is very costly for the female. If there were negative effects on egg production, one would also expect to find costs during incubation when, if moulting has already started, moulting would be more resource intensive.

As it was stressed above, parents were captured and their moulting state assessed when their nestlings were 10–12 days old. Considering 7 days for laying eggs, 13 days for incubation and 10 days of the nestling period until capture of the parents, this means that the moult state was assessed about one month after the clutch initiation date. As it is the case that the most advanced birds were moulting their fourth primary when captured, and many of them were still moulting their first or second ones, the moulting process of most of them could not have been started, or was in its very early phase, when the clutch was produced. Thus, the lack of negative effects attributable to the overlap could be simply due to a lack of an extensive overlap at this time. Contrasting with this, parents found moulting when captured clearly overlapped this activity with the feeding of their young, so negative consequences on the number and condition of fledglings would be more probable to occur. We do not have any direct comparison to perform, since the other two observational studies available did not directly consider the number of fledglings produced. Hemborg et al., (2001) [26] found that breeding success (proportion of eggs producing fledglings) did not differ between moulting and non-moulting females in any of four studied European pied flycatcher populations, while it was lower for moulting males. The experimental studies, on the other hand, do not provide a clear guide, since two of them [20,23] found a decrease in the number of fledglings, while the other two failed to detect this effect [19,24]. Each study had a different experimental protocol, for instance, delaying breeding time, plucking feathers to simulate moult, providing extra food, altering brood size, etc., at different moments of the breeding cycle. Therefore, the effects of overlapping could be misleaded with the effects of manipulation. For example, Hemborg (1998) [20] found that delayed pairs (supposedly overlapping more frequently) produced fewer fledglings, but the moult status of the birds did not affect fledgling number. Our results are in full agreement with this conclusion: the lower number of fledglings produced by overlapping Spanish great tits is mainly due to the fact that they breed later in the season, not because they are moulting.

Some experimental studies have shown that overlapping birds produced lighter fledglings or fledglings in poorer body condition [10,21,22]. Similar to our results on other breeding parameters, nestling body condition did not differ between moulting and non-moulting pairs once the laying date was taken into account.

## 5. Conclusions

In summary, we have observed that the overlap of breeding and moulting does not impose immediate reproductive costs on birds that moult “voluntarily” (i.e., are not induced to moult) while breeding. Obviously, these costs could manifest themselves later, as differential survival of fledglings and/or as differential survival and/or future reproductive costs for adults. This has been shown in some studies, both experimental [10,24] and observational [17,25,27], but not in others [19,20,21,22,23].

## Figures and Tables

**Figure 1 animals-13-00409-f001:**
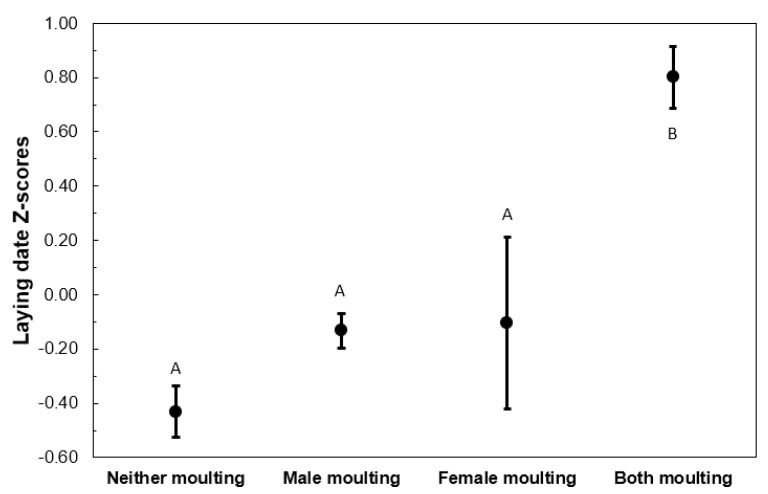
Residuals of the laying date of the second clutches of pairs that could potentially have overlapped moult and breeding, as a function of the moult status of the two members of the pair. Letters above the bars show differences between groups according to Tukey’s HSD tests. Sample size = 334 pairs.

**Figure 2 animals-13-00409-f002:**
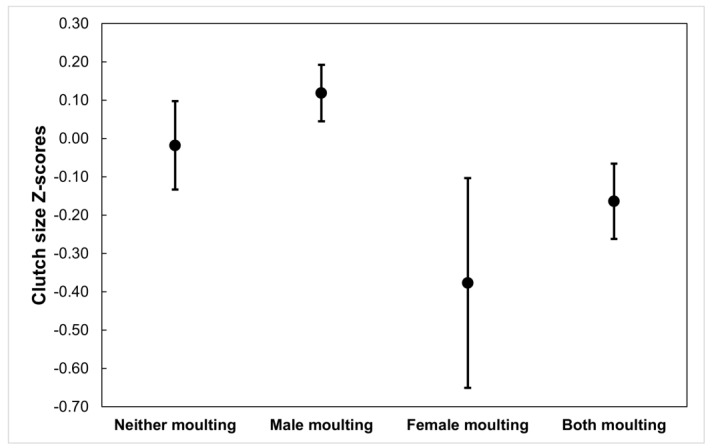
Residuals of the size of the second clutch of pairs that could potentially have overlapped moult and breeding, after removing the effects of laying date, as a function of the moult status of the two members of the pair. Sample size = 332 pairs.

**Figure 3 animals-13-00409-f003:**
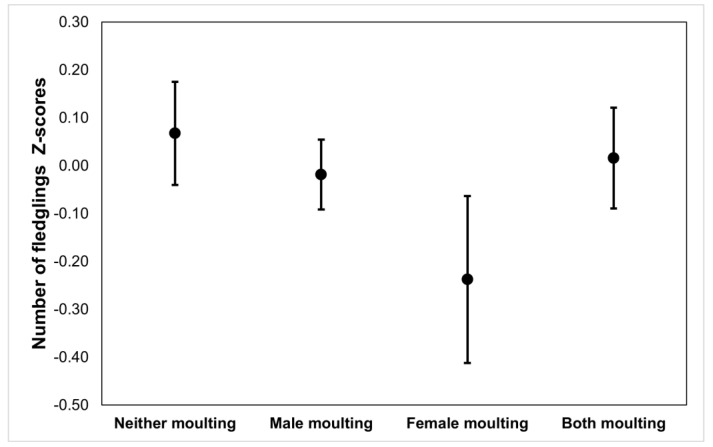
Residuals of the number of fledglings produced in their second clutch by pairs that could potentially have overlapped moult and breeding, after removing the effects of laying date, and as a function of the moult status of the two members of the pair. Sample size = 332 pairs.

## Data Availability

The data presented in this study are openly available in Zenodo at DOI, 10.5281/zenodo.7398652.

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
