# Peer review of "No Apparent Immediate Reproductive Costs of Overlapping Breeding and Moult in a Mediterranean Great Tit Population"

_animals, 2023, doi:10.3390/ani13030409_

Round 1

Reviewer 1 Report

This manuscript studies the breeding and molting overlapping of the great tit population, and trying to explain whether the overlapping of the breeding and molting has an impact on the reproductive costs. In general, the research design of this paper is more reasonable and the method is appropriate, which is a interesting study.

 1. In the introduction section, the author should add information about the past breeding and molting time of the great tit population studied in this paper. The author puts forward that the premise of this study is that climate change affects the breeding and molting time of birds, but in the introduction, I do not see whether the breeding and molting time of this population is affected by climate change, or that the overlapping of breeding and molting time described later in this paper is always present in the local great tit population.

 2. In addition, the author did not explain whether moulting during breeding period has an impact on the rearing of offspring by parent birds. For example, some migratory birds cannot fly long distances when moulting is expected, and whether moulting of great tit has a similar impact is not described in the text.

 3. In the introduction part, the author cited too few relevant literatures on the study of great tit molting, and I suggested to supplement them.

 4. The study lasted from 1995 to 2019. During the 25 years, the author should supplement whether the habitat of great tits in the study area has changed, such as whether the vegetation and food resources have changed. This is also an important factor affecting the reproduction and molting of great tits. Otherwise, the persuasiveness of the author's analysis conclusion will be greatly reduced.

 5. In the research result 3.1. Incidence of Moult breeding Overlap, the author should clarify that all the breeding in this study is the second breeding, and no molting is observed in all the first breeding, which is helpful for readers to understand the whole impact process, which is very important information.

 6. Clutch size, the number of fledglings produced, the nestlings, all these will be affected by many factors, such as breeding time, experience of breeding pairs, abundance of food resources during breeding, etc. However, the study of Overlapping Breeding and Moult in this paper is not a direct external factor. So, in the discussion section, relevant information should be supplemented to illustrate that these factors are also considered in this study.

Author Response

RESPONSE TO REVIEWER 1 COMMENTS:

This manuscript studies the breeding and molting overlapping of the great tit population, and trying to explain whether the overlapping of the breeding and molting has an impact on the reproductive costs. In general, the research design of this paper is more reasonable and the method is appropriate, which is a interesting study.

  1. In the introduction section, the author should add information about the past breeding and molting time of the great tit population studied in this paper. The author puts forward that the premise of this study is that climate change affects the breeding and molting time of birds, but in the introduction, I do not see whether the breeding and molting time of this population is affected by climate change, or that the overlapping of breeding and molting time described later in this paper is always present in the local great tit population.

Response 1: Both the advancement of the breeding season and its relation with global warming (Solís et al. 2022), and the increase of the moult-breeding overlap in males and females of this population related to this advancement of laying dates (Solís et al. 2021) have been previously reported in already published papers. We have emphasized both aspects in the last paragraph of the Introduction, and in the first one of the Materials and Methods section.

  1. In addition, the author did not explain whether moulting during breeding period has an impact on the rearing of offspring by parent birds. For example, some migratory birds cannot fly long distances when moulting is expected, and whether moulting of great tit has a similar impact is not described in the text.

Response 2: We added some more information and references for this question along the main text and explicitly in the second paragraph of the Introduction. Great tits, to our knowledge, as all passerines, perform a sequential moult of feathers, so the flying performance is somewhat reduced, but they are able to develop their activities. The main problem for them is probably energy allocation when two energy-demanding processes compete for resources.

  1. In the introduction part, the author cited too few relevant literatures on the study of great tit molting, and I suggested to supplement them.

Response 3: We are citing probably all the studies reporting effects of moult-breeding overlap on reproductive performance. None of them was performed on great tits. This is the target of the study. The moult of great tits itself is not dealt with in this study, and a complete list of references on great tit moult is provided in a previous paper (Solís et al. 2021). We do not find necessary to deepen on this topic here.

  1. The study lasted from 1995 to 2019. During the 25 years, the author should supplement whether the habitat of great tits in the study area has changed, such as whether the vegetation and food resources have changed. This is also an important factor affecting the reproduction and molting of great tits. Otherwise, the persuasiveness of the author's analysis conclusion will be greatly reduced.

Response 4: This is now reported on the first paragraph of the Materials and Methods section.

  1. In the research result 3.1. Incidence of Moult breeding Overlap, the author should clarify that all the breeding in this study is the second breeding, and no molting is observed in all the first breeding, which is helpful for readers to understand the whole impact process, which is very important information.

Response 5: Done – see third paragraph of the Materials and Methods section.

  1. Clutch size, the number of fledglings produced, the nestlings, all these will be affected by many factors, such as breeding time, experience of breeding pairs, abundance of food resources during breeding, etc. However, the study of Overlapping Breeding and Moult in this paper is not a direct external factor. So, in the discussion section, relevant information should be supplemented to illustrate that these factors are also considered in this study.

Response 6: We have standardized these dependent variables by year, thus removing between year variation in weather or food availability and considered the laying dates into the analyses (in fact, this is the main factor explaining differences in breeding performance). The characteristics of the birds (age/experience) were not considered because we took complete pairs as sample units – it would probably be impossible, with the available sample size, to also consider all the possible combinations of age-classes within each “moult-type” pair.

Reviewer 2 Report

This is an interesting study that explore the increasingly occurred overlap between breeding and moulting on breeding performance of a non-migrant bird, great tit. The results indicate that, after taking into account laying dates, reproductive parameters, including clutch size, number of hatchlings, number of fledglings, and body condition of nestlings, were not negatively affected by overlapping breeding and moult. This study can improve our understanding of the impacts of global climate warming on breeding and fitness of animals.

 I only have one comment. This study was based on reproductive data of 390 pairs of great tits collected between 25 years (i.e., 1995-2019). The authors integrated all data and analyzed them together, without considering the potential influence of environmental factors, especially temperature, which I guess temperature might increase within the long studying periods. How do you think about the influence of temperature on reproductive performance? I suggest you consider this factor when analyze your data.

Author Response

RESPONSE TO REVIEWER 2 COMMENTS:

This is an interesting study that explore the increasingly occurred overlap between breeding and moulting on breeding performance of a non-migrant bird, great tit. The results indicate that, after taking into account laying dates, reproductive parameters, including clutch size, number of hatchlings, number of fledglings, and body condition of nestlings, were not negatively affected by overlapping breeding and moult. This study can improve our understanding of the impacts of global climate warming on breeding and fitness of animals.

I only have one comment. This study was based on reproductive data of 390 pairs of great tits collected between 25 years (i.e., 1995-2019). The authors integrated all data and analyzed them together, without considering the potential influence of environmental factors, especially temperature, which I guess temperature might increase within the long studying periods. How do you think about the influence of temperature on reproductive performance? I suggest you consider this factor when analyze your data.

Response: The advancement of the breeding season and its relation with global warming has been previously reported (Solís et al. 2022). We are comparing here the reproductive performance of pairs breeding at the same time (some overlapping and some others not). We know there have been changes in the breeding performance along the study period, but this is not the target of this particular study.

Reviewer 3 Report

Manuscript ID: animals-2115633

Title: No Apparent Immediate Reproductive Costs of Overlapping Breeding and Moult in a Mediterranean Great Tit Population

General comments

This is a very interesting manuscript, concerning a remarkable argument, well written and easy to read. Nonetheless, I found some issues to be addressed, regarding the clarity of methods (see specific comments). I have only a major suggestion. You have collected data for many years, therefore, would be interesting to see if there was a temporal trend in the % of the four “moulting groups” since 1995. And if there was a trend, should be interesting to relate it with temperature. I repeat, this is only a suggestion, but it would be nice to see these analyses to contextualize better.

Specific comments

Materials and methods

·       Line 94: how many nest-boxes? what’s their density inside the groves?

·       Line 97: please clarify ‘in some period’

·       Line 106: please add some references for using this method (ratio between body mass and tarsus length)

·       Lines 107-109: are you sure that the absence of nestlings is attributable to fledging and not predation? Please clarify this concept.

Author Response

RESPONSE TO REVIEWER 3 COMMENTS:

General comments

This is a very interesting manuscript, concerning a remarkable argument, well written and easy to read. Nonetheless, I found some issues to be addressed, regarding the clarity of methods (see specific comments). I have only a major suggestion. You have collected data for many years, therefore, would be interesting to see if there was a temporal trend in the % of the four “moulting groups” since 1995. And if there was a trend, should be interesting to relate it with temperature. I repeat, this is only a suggestion, but it would be nice to see these analyses to contextualize better.

Response: This is an interesting suggestion, but changes in the proportion of moulting pairs and individuals are not the target of the present study. We explored this in a previous work (Solís et al. 2021). Nevertheless, just for curiosity, we have done the analyses. We have not found any significant temporal trend in the percentages in pairs where none was moulting (R2 = 0.1%, F1,24 = 0.016, P = 0.901), where only the male was moulting (R2 = 8.7%, F1,24 = 2.183, P = 0.153) and where only the female was moulting (R2 = 6%, F1,24 = 1.471, P = 0.237). Pairs where both individuals were moulting increased over time (R2 = 61.3%, F1,24 = 36.493, P < 0.001). This result is consistent with that reported by Solís et al. (2021). We do not think this is relevant, however, for the present study, where the target is, once we know that some individuals are moulting, finding out the reproductive consequences. This has not been included in the revised version, but we could do so if the editor considers it relevant.

Specific comments

Materials and methods

  • Line 94: how many nest-boxes? what’s their density inside the groves?

Response: Now reported in the first paragraph of the Materials and Methods section.

  • Line 97: please clarify ‘in some period’

Response: Now reported in the second paragraph of the Materials and Methods section.

  • Line 106: please add some references for using this method (ratio between body mass and tarsus length)

Response: Added at the second paragraph of Materials and Methods section.

  • Lines 107-109: are you sure that the absence of nestlings is attributable to fledging and not predation? Please clarify this concept.

Response: We have explained this at the third paragraph of the Materials and Methods section.